# Expression, not sequence, distinguishes miR-238 from its miR-239ab sister miRNAs in promoting longevity in *Caenorhabditis elegans*

**Laura B. Chipman**[¤], **San Luc**, **Ian A. Nicastro, Jesse J. Hulahan, Delaney C. Dann**, **Devavrat M. Bodas, Amy E. Pasquinelli***

Molecular Biology Department, School of Biological Sciences, University of California, San Diego, La Jolla, California, United States of America

¤ Current address: Center for Therapeutic Innovation, Pfizer Inc., 610 Main Street, Cambridge, MA 02139, United States of America
* apasquinelli@ucsd.edu

**Data Availability Statement:** All RNA Sequencing data files are available from the GEO database (https://www.ncbi.nlm.nih.gov/geo/query/acc.cgi?

## Abstract

MicroRNAs (miRNAs) regulate gene expression by base-pairing to target sequences in messenger RNAs (mRNAs) and recruiting factors that induce translational repression and mRNA decay. In animals, nucleotides 2–8 at the 5' end of the miRNA, called the seed region, are often necessary and sometimes sufficient for functional target interactions. MiRNAs that contain identical seed sequences are grouped into families where individual members have the potential to share targets and act redundantly. A rare exception seemed to be the miR-238/239ab family in *Caenorhabditis elegans*, as previous work indicated that loss of miR-238 reduced lifespan while deletion of the *miR-239ab* locus resulted in enhanced longevity and thermal stress resistance. Here, we re-examined these potentially opposing roles using new strains that individually disrupt each miRNA sister. We confirmed that loss of miR-238 is associated with a shortened lifespan but could detect no longevity or stress phenotypes in animals lacking miR-239a or miR-239b, individually or in combination. Additionally, dozens of genes were mis-regulated in *miR-238* mutants but almost no gene expression changes were detected in either *miR-239a* or *miR-239b* mutants compared to wild type animals. We present evidence that the lack of redundancy between *miR-238* and *miR-239ab* is independent of their sequence differences; miR-239a or miR-239b could substitute for the longevity role of miR-238 when expressed from the *miR-238* locus. Altogether, these studies disqualify miR-239ab as negative regulators of aging and demonstrate that expression, not sequence, dictates the specific role of miR-238 in promoting longevity.

## Author summary

MicroRNAs (miRNAs) are tiny non-coding RNAs that function in diverse biological pathways. To exert their regulatory influence, miRNAs bind to specific target RNAs through partial base-pairing. A critical aspect of this miRNA-target engagement is the seed sequence, nucleotides 2–8 of the miRNA. MiRNAs that share seed sequences are

acc=GSE232471) with accession number GSE232471.

**Funding:** Support for this study was provided by a UCSD Cellular and Molecular Genetics Training Program institutional grant from National Institute of General Medical Sciences (https://www.nigms.nih.gov/, T32 GM007240 to LBC and DCD), an NSF Graduate Research Fellowship (https://www.nsfgrfp.org, DGE-1650112 to LBC) and a University of California, San Diego Eureka! Scholarship (DMB). This work was funded by grants from the NIH through NIGMS (https://www.nigms.nih.gov/) GM127012 and NIA (https://www.nia.nih.gov) AG056562, and AFAR (https://www.afar.org) to AEP. Some strains were provided by the C. elegans Genetics Center (CGC) (https://cgc.umn.edu), which is funded by NIH Office of Research Infrastructure Programs (P40 OD010440) (https://orip.nih.gov). Some data were generated at the UC San Diego IGM Genomics Center utilizing an Illumina NovaSeq 6000 that was purchased with funding from a National Institutes of Health SIG grant (https://www.nih.gov) (#S10 OD026929). The funders had no role in study design, data collection and analysis, decision to publish, or preparation of the manuscript.

**Competing interests:** The authors have declared that no competing interests exist.

grouped into families and presumed to have similar functions. Yet, other factors, such as non-seed sequences in the miRNA and its expression level, can also contribute to target regulation and result in distinct roles for miRNAs within a family. To better understand how miRNA family members can have specific functions, we focused on miR-238 and its sisters, miR-239a and miR-239b, because these miRNAs had previously been reported to play opposing longevity roles in the nematode *C. elegans*. Using new genetic tools, we found that loss of miR-238 alone leads to the misregulation of many genes and a reduced lifespan. However, the lack of miR-239a, miR-239b, or both sisters had almost no effect on gene expression or longevity compared to wild type animals. Strikingly, though, miR-239a or miR-239b could substitute for the aging role of miR-238 when expressed from the miR-238 locus. Thus, expression, not sequence, is the predominant distinguishing feature of mir-238 that bestows upon it a role in aging not shared with the other family members.

## Introduction

MicroRNAs (miRNAs) are small, ~22 nucleotide (nt), RNA regulators that post-transcriptionally repress target RNAs in a sequence dependent manner [1–3]. Most metazoan miRNAs are transcribed into long primary miRNAs (pri-miRNAs) by RNA Pol II with a stem-loop structure that is recognized and processed into a ~60 nt precursor miRNA (pre-miRNA) [4,5]. Dicer cuts both strands of the pre-miRNA stem-loop structure, leaving a miRNA duplex where one strand will be degraded and the other bound by an Argonaute (AGO) protein [2,6]. Once a miRNA is loaded into AGO, it forms the core of the miRNA induced silencing complex (miRISC), which induces translational inhibition and decay of the target RNA [2]. While animal miRNAs typically use partial base-pairing to engage a target sequence, perfect complementarity with miRNA nucleotides 2–8, called the "seed", is a predominant feature of target recognition [2,7]. Structural studies have revealed that the nucleotides available for initial target recognition and base-pairing are limited to the seed region in miRNAs bound by Argonaute [8,9]. Due to the importance of the miRNA seed sequence in targeting as well as being the most evolutionarily conserved region, miRNAs that share a seed sequence are grouped into families [2]. Given the reliance of targeting on the seed sequence, it is often assumed that miRNA family members function redundantly. This idea is supported by numerous studies showing that loss of entire miRNA families, and not individual members, is often required for phenotypic consequences [10–13].

Yet, recent work has highlighted that sequences beyond the seed, as well as miRNA expression levels, can also play roles in determining functional miRNA-target interactions [2,14]. High-throughput capture assays of miRNA-target complexes have revealed a high frequency of interactions with partial or poor seed matches between the miRNA and its target RNA, some with extensive base-pairing to the 3' end of the miRNA [15–19]. Biochemical and structural studies have shown that increased 3' end pairing can strengthen miRNA repression by increasing miRNA-target affinity [20–23]. As well, *in vivo* studies in *C. elegans* have shown that pairing of the 3' region of the miRNA can facilitate miRNA-target interactions and confer target specificity among miRNA family members who share their seed sequence but differ in their 3' ends [15,24–26]. Furthermore, increases in miRNA concentration can sometimes compensate for suboptimal pairing architectures [24]. At the same time, there is evidence that sequences in the 3' half of some miRNAs are irrelevant for *in vivo* functions [12,27]. Thus, the role of individual miRNA family members that differ in 3' end sequences, expression levels, and locations can be unpredictable.

A particularly intriguing miRNA family is the miR-238/239ab family in *C. elegans*. The miR-238, miR-239a, and miR-239b miRNAs are identical from nucleotides 1–8 and then diverge, with maintenance of a high degree of homology between the miR-239a and miR-239b 3' ends. Moreover, strains that lack miR-238 or miR-239ab were reported to exhibit opposite longevity phenotypes [28]. While *miR-238(-)* had a reduced lifespan, a strain deleted of *miR-239a*, *miR-239b*, and their surrounding genomic sequences (*nDf62)* showed an extended lifespan [28]. Additionally, the *nDf62* strain displayed enhanced resistance to heat and oxidative stress, whereas *miR-238* mutants were more sensitive to oxidative stress and survived at a rate comparable to that of wildtype worms in a thermal stress assay [28]. Thus, the miR-238/239ab family has been considered an unusual example of related miRNAs with opposing roles in longevity and stress pathways.

In this study, we set out to better define the contribution of the individual miR-238/239ab family members to the regulation of lifespan and thermal resistance in *C. elegans*. Consistent with previous work [28], we found that deletion of *miR-238* alone resulted in a shortened lifespan. However, neither individual nor combined loss of miR-239a and miR-239b produced the enhanced longevity or heat stress tolerance previously attributed to deletion of these miRNAs. The single *miR-239a* and *miR-239b* mutant strains also displayed almost no changes in gene expression compared to wildtype animals, whereas ~70 genes were mis-regulated in the *miR-238* mutants. In addition to divergence in 3' end sequences, differences in expression could underlie the inability of miR-239ab to compensate for the loss of miR-238. Consistent with the latter possibility, we found that rescue of the reduced lifespan caused by loss of miR-238 was achieved by replacing the precursor for miR-238 with that of miR-239a or miR-239b at the endogenous *miR-238* locus. Altogether, our data reveal that the miR-238/239ab family plays a positive role in aging that is primarily reliant on expression and independent of sequence differences among the miRNAs.

## Results

### Levels of miR-238, miR-239a, or miR-239b are minimally perturbed by loss of other family members

As members of the same miRNA family in *C. elegans*, the miR-238, miR-239a, and miR-239b miRNAs share their seed sequence, nucleotides 2–8, but differ to varying degrees in their 3' ends (Fig 1A–1C). The previous deletion strain (*nDf62*) used to characterize the function of miR-239a and miR-239b removes both miRNA sisters, as well as a ncRNA and snoRNA (Fig 1A) [13]. This is unlike the *miR-238(n4112)* allele, which disrupts the *miR-238* gene and no other annotated genes in the vicinity (Fig 1B) [13]. To study the contribution of the individual miRNA sisters, miR-239a and miR-239b, to aging phenotypes, we used CRISPR/Cas9 to create new, single loss of function (LOF) alleles. Due to the high sequence similarity within the mature miR-238, miR-239a, and miR-239b sequences, we targeted the pre-miRNA to disrupt miRNA processing and, thus, mature miRNA levels. Using this strategy, we made a new LOF allele for miR-239a, *miR-239a(ap439)*, two new LOF alleles for miR-239b, *miR-239b(ap432)* and *miR-239b(ap433)*, and a miR-239a and miR-239b dual LOF strain *miR-239a/b(ap435, ap432)* (Fig 1A). The mutations altered the predicted structures of the miRNA precursors by either inserting (*ap435*), deleting (*ap432*, ap439) or inserting and deleting (ap433) sequences within the miRNA genes (Figs 1A and S1) [29]. These disruptions prevented the accumulation of mature miRNAs, as we were unable to detect miR-239a or miR-239b in their corresponding mutant backgrounds (Fig 1D). In the dual LOF strain *miR-239a/b(ap435,ap432)*, miR-239a was barely detectable at levels less than 1% of WT expression, indicating that there is drastically impaired production of miR-239a from the *ap435* allele (Fig 1D). To test if these disruptions

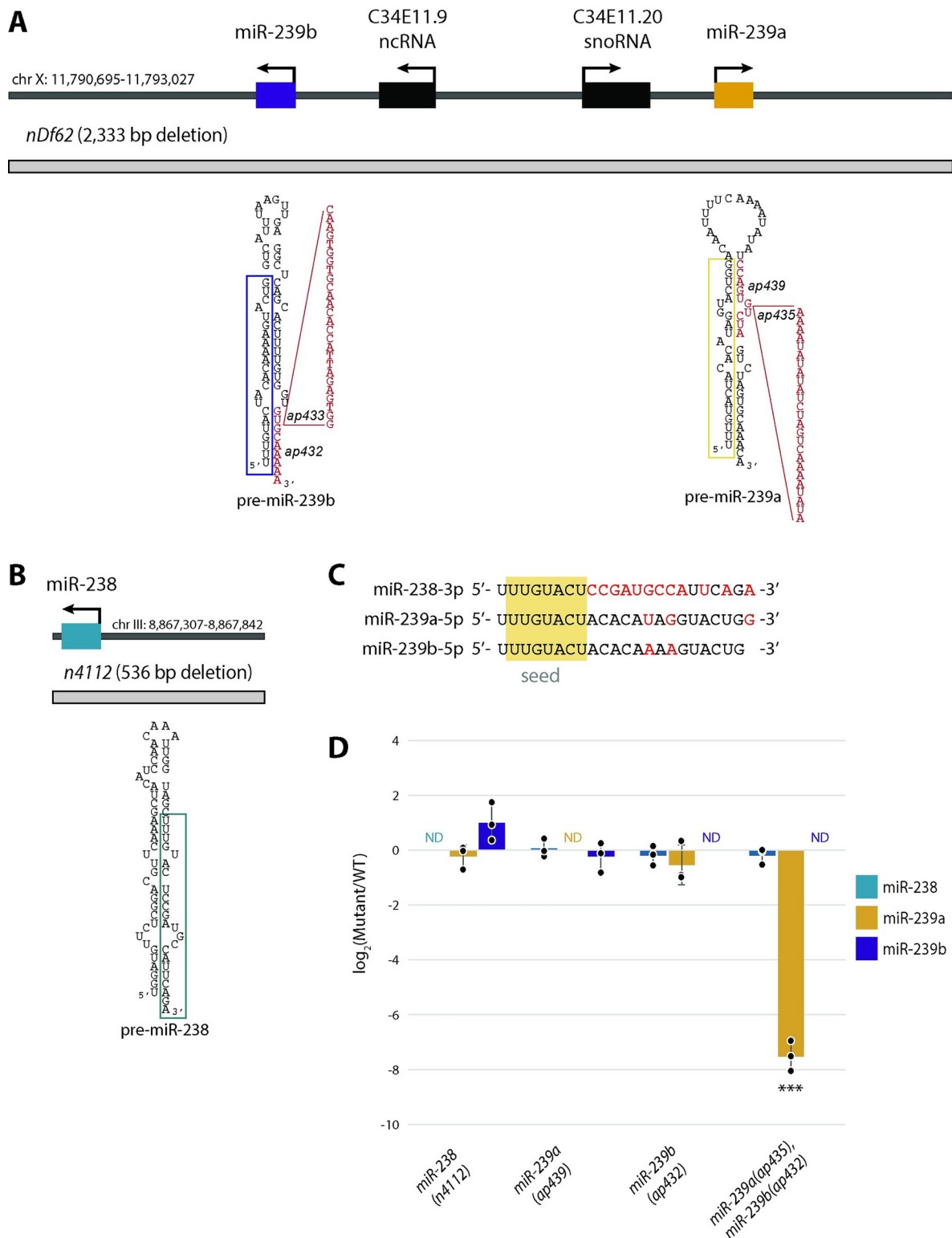

**Fig 1. Validation of individual *miR-238, miR-239a, miR-239b* loss of function strains.** (A-B) The genomic loci of *miR-239b* and *miR-239a* (A), and *miR-238* (B) with surrounding genomic features. Gene directionality is indicated with black arrows. The gray boxes mark regions deleted in the *miR-239ab (nDf62)* and *miR-238(n4112)* strains. New loss of function mutants generated in this study by CRISPR/Cas9 are indicated in red in the precursor structures; mature sequences are boxed. *miR-239b(ap432)* deletes 15 nt at the base of the 3' arm of the stem (not all nucleotides are shown) and *miR-239b(ap433)* deletes the GCAAAAA sequence and inserts 26 nt. *miR-239a(ap439)* deletes

10 nucleotides in the 3' arm of the stem and *miR-239a(ap435)* inserts 25 nucleotides into this region. (C) The miR-238-3p, miR-239a-5p and miR-239b-5p miRNAs share their seed sequence, nucleotides 2–8 (shaded gold), but differ in other sequences, indicated in red. (D) TaqMan RT-qPCR analysis of miR-238, miR-239a, miR-239b mature miRNA levels in WT, *miR-238(n4112)*, *miR-239a(ap439)*, *miR-239b(ap432)*, *miR-239a/b(ap435,ap432)* L4 stage animals. The mean from 3 independent replicates is plotted; error bars represent SDs, dots represent individual replicates, ND = Not Detected. Statistical significance assessed by student's two-tail t-test, *** P<0.0001.

in miR-238, miR-239a, or miR-239b mature miRNA production led to altered expression of the other sisters, we examined the mature miRNA levels of each sister in *miR-238(n4112)*, *miR-239a(ap439)*, *miR-239b(ap432)* individual LOF strains as well as in the double mutant, *miR-239a/b(ap435,ap432)*. Little if any change was detected for any of the miRNA sisters upon deletion of one or two members of its family (Fig 1D). These results confirm that we have valid new tools to assess how loss of individual miR-238/239ab miRNA sisters contributes to aging.

## Loss of miR-238 leads to a reduced lifespan, while loss of miR-239a or miR-239b has no effect on *C. elegans* lifespan

As seen in previous work, we observed that loss of *miR-238(n4112)* resulted in a reduced lifespan, implicating it as a positive regulator of longevity (Fig 2A) [28]. In contrast to the previously published extended lifespan attributed to loss of miR-239ab in the *nDf62* strain [28], individual or coupled loss of miR-239a and miR-239b did not significantly alter lifespan compared to WT (Fig 2A and 2B). Furthermore, a strain lacking expression of the entire miR-238/239ab family had a similarly reduced lifespan as loss of miR-238 alone (Fig 2A). Together, these data suggest that miR-238 plays an important role in aging adults, while miR-239a and miR-239b have no influence on lifespan.

## The miR-238/239ab family is nonessential for fertility and heat stress recovery in early adulthood

The reduced lifespan of *miR-238(n4112)* is not apparently linked to any obvious developmental or other defects [13,28]. We also found that the loss of miR-238 or miR-239b had no significant effect on fertility, as judged by brood size analysis (Fig 2C and 2D). While the *miR-239a (ap439)* mutants produced slightly fewer progeny than WT animals, this difference was not observed in the double *miR-239a(ap435)*, *miR-239b(ap432)* or triple *miR-238(n4112); miR-239a(ap435)*, *miR-239b(ap432)* loss of function strains (Fig 2C and 2D). Overall, the miR-238/239ab family seems to have a minor, if any, role in development and fertility under typical laboratory conditions.

The miR-238/239ab family has also been reported to differentially regulate stress responses. Previously, the *miR-239a/b(nDf62)* strain was shown to have increased thermotolerance and thermoresistance in adults [28,30]. While the *miR-238(n4112)* strain did not exhibit a heat shock phenotype, it was more sensitive to oxidative stress, and, conversely, *miR-239a/b(nDf62)* animals were more resistant to this stress than WT [28]. When we attempted to recapitulate the thermotolerance assay, which subjected day 2 adults to 12hr of heat shock at 35°C [28], all animals died. However, a thermoresistance assay, where day 2 adults were exposed to 15hr of heat shock at 32°C and scored for survival after a 24hr recovery period at 20°C, resulted in survival of WT animals at levels previously observed for this assay (Fig 2E and 2F) [30]. Although all the individual and combined mutant strains trended towards lower survival rates compared to WT, there was no statistically significant difference (Fig 2E and 2F). Taken together, the miR-238/239ab family does not substantially contribute to thermoresistance, as assayed here in adult *C. elegans*.

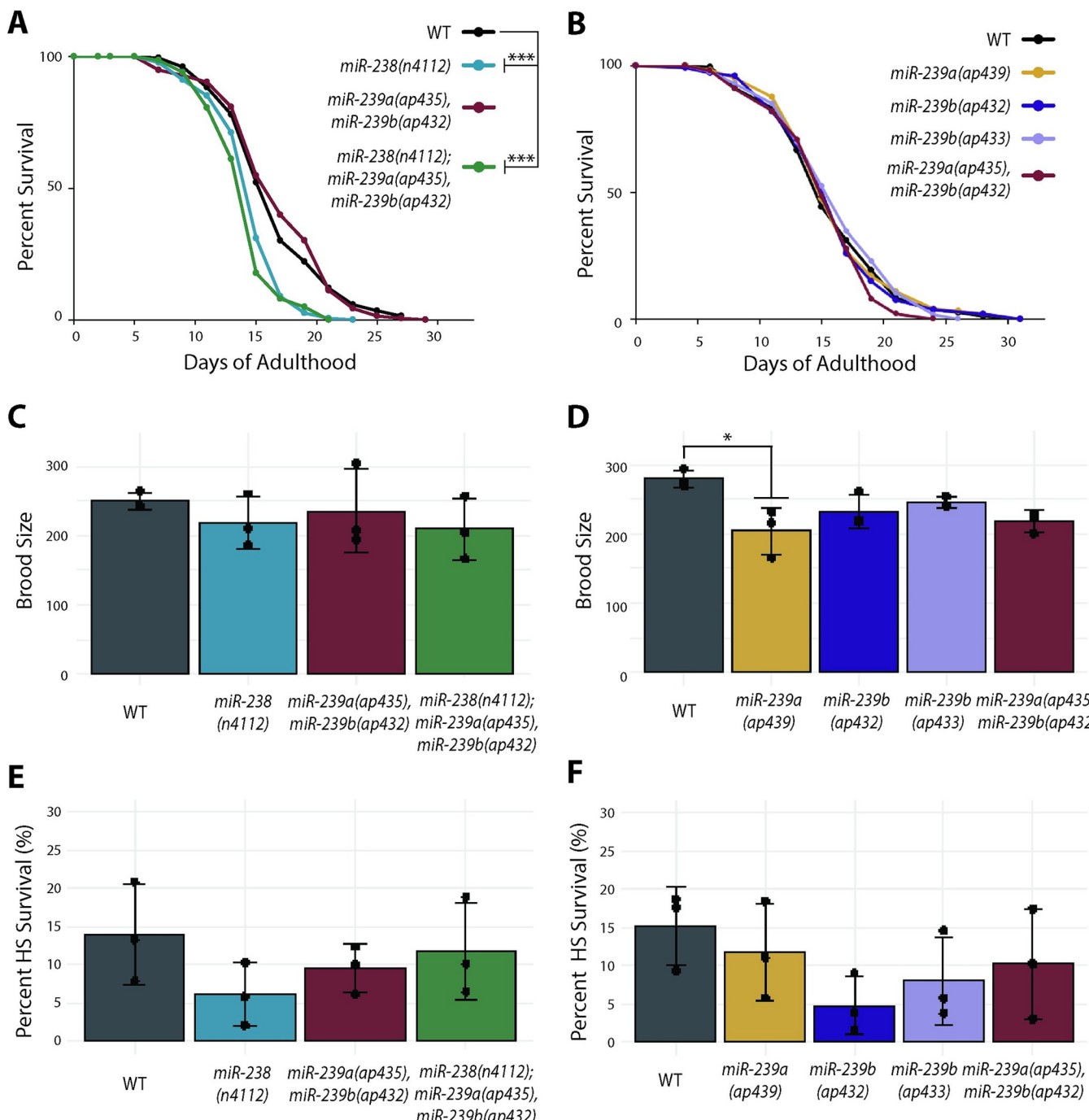

**Fig 2. The miR-238/239a/239b sisters have distinct roles in adult *C. elegans*.** (A) Representative survival curves for WT (black), *miR-238(n4112)* (aqua), *miR-239a/b(ap435,ap432)* (maroon), and *miR-238(n4112);miR-239a/b(ap435,ap432)* (green) that show a reduced lifespan of *miR-238(n4112)* (aqua), and *miR-238(n4112);miR-239a/b(ap435,ap432)* (green) compared to WT (black). *** P<0.0001 (log-rank). (B) Representative survival curves for WT (black), *miR-239a (ap439)* (gold), *miR-239b(ap432)* (purple), *miR-239b(ap433)* (light purple), *miR-239a/b(ap435,ap432)* (maroon). No significant difference in lifespan when compared to WT. (C-D) Results from brood size analysis. Bar graph represents mean of three biological replicates, individual replicate data are indicated with black dots. The error bars represent SDs. (C) *MiR-238(n4112)* (aqua), *miR-239a/b(ap435,ap432)* (maroon), and *miR-238(n4112);miR-239a/b(ap435,ap432)* (green) do not have a statistically significant difference when compared to WT (black). ANOVA and the post hoc test (Tukey's HSD). (D) *MiR-239a(ap439)* (gold), *miR-239b(ap432)* (purple), *miR-239b(ap433)* (light purple), *miR-239a/b(ap435,ap432)* (maroon). *P<0.05, ANOVA and the post hoc test (Tukey's HSD). (E-F) Results from heat shock assay of day 2 adults for 15 hours at 32°C followed by recovery for 24 hours at 20°C. Bar graph represents mean of three biological replicates, individual replicate data indicated with black dots. The error bars represent SDs. (E) *MiR-238(n4112)* (aqua), *miR-239a/b(ap435,ap432)* (maroon), and *miR-238(n4112);miR-239a/b(ap435,ap432)* (green) do not have a statistically significant difference when compared to WT (black). ANOVA and

the post hoc test (Tukey's HSD). (F) *MiR-239a(ap439)* (gold), *miR-239b(ap432)* (purple), *miR-239b(ap433)* (light purple), *miR-239a/b(ap435,ap432)* (maroon) do not have a statistically significant difference when compared to WT (black). ANOVA and the post hoc test (Tukey's HSD).

## Non-overlapping sets of genes are mis-regulated upon loss of each miR-238/239ab family member

Given that miRNAs are post-transcriptional gene regulators that often induce degradation of their target mRNAs [31], we asked if similar sets of genes would be mis-regulated upon loss of each miR-238/239ab family member. We performed transcriptomic analysis on day 5 adult animals of the individual *miR-238*, *miR-239a*, and *miR-239b* mutants, along with WT for comparison (S1 Table). In the *miR-238(n4112)* mutants, there was significant (padj < 0.05 and baseMean >100) up-regulation of 25 genes and down-regulation of 44 genes (Fig 3A and Table 1). In contrast, for the *miR-239a (ap439)* and *miR-239b(ap432)* loss of function mutants, very few genes were found to be differentially expressed compared to WT (Fig 3B and 3C). The single gene mis-regulated in the *miR-239a* mutant is *C34E11.20*, a snoRNA adjacent to *miR-239a* (Figs 1A and 3B). While the *ap439* deletion does not span the annotated *C34E11.20* gene locus, the genomic disruption, rather than the loss of miR-239a, is likely responsible for the altered expression of this snoRNA. None of the mis-regulated genes in the three mutant backgrounds has a miR-238/239 binding site predicted by TargetScan [32,33], suggesting that the change in mRNA levels is an indirect consequence of loss of the miRNAs. Although it is possible that the adult day 5 time point missed gene expression changes of direct targets or these miRNAs primarily cause translational repression without substantial target mRNA degradation at this time point in adulthood, the lack of over-lapping downstream effects suggests that miR-238, miR-239a and miR-239b mostly regulate different genes. Furthermore, these data reflect the lifespan phenotypes with loss of miR-238 resulting in a greater extent of gene mis-regulation and a reduced lifespan and loss of miR-239a or miR-239b having almost no effect on gene expression and longevity.

## Members of the miR-238/239ab miRNA family are differentially expressed

Members of the miR-238/239ab family were originally identified as potential regulators of longevity due to their increase in expression during aging [28]. To further study the levels of these miRNAs in adult animals, we performed small RNA transcriptomics on adult day 5 wildtype (WT) animals (S2 Table). Along with a previously generated small RNA-seq (smRNA-seq) dataset from larval stage 4 (L4) WT worms [39], we ranked the expression of miR-238, miR-239a and miR-239b relative to all other detected miRNAs (Fig 4A). From these rankings, miR-238 is the most abundantly detected family member in L4 and day 5 (Fig 4A). Additionally, miR-238 has a slight increase in ranking from L4 to day 5 (Fig 4A). Both miR-239a and miR-239b increase in ranking ~2 fold from L4 to day 5 but still are detected less frequently than miR-238 (Fig 4A). Given that smRNA-seq can be subject to ligation bias, thus leading to uneven quantification of miRNAs of different sequences [40], we also performed absolute quantification using RNA oligo standards. These analyses indicate that miR-238 levels are approximately 6-fold higher than those of miR-239b at day 5 of adulthood (Fig 4B), a difference comparable to the small RNA-seq results (S2 Table).

Previous studies have also noted differences in the spatial expression patterns of miR-238 and miR-239ab [28,42,43]. We made similar observations when we examined GFP expression driven by the promoters, defined as ~2kb of sequence upstream of the mature miRNA sequence, of miR-238 and miR-239b in adult day 5 animals. The $_p$miR-238::GFP reporter was transcribed nearly ubiquitously, with highest levels detected in the intestine, hypodermis, and

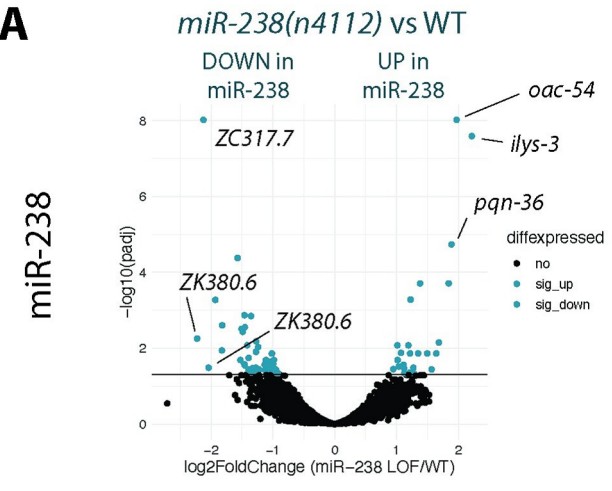

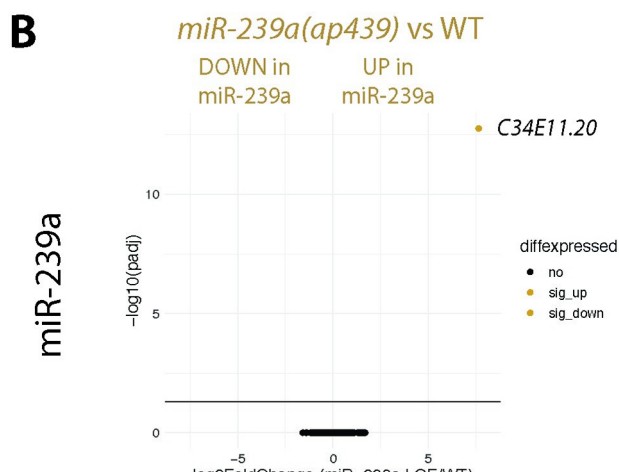

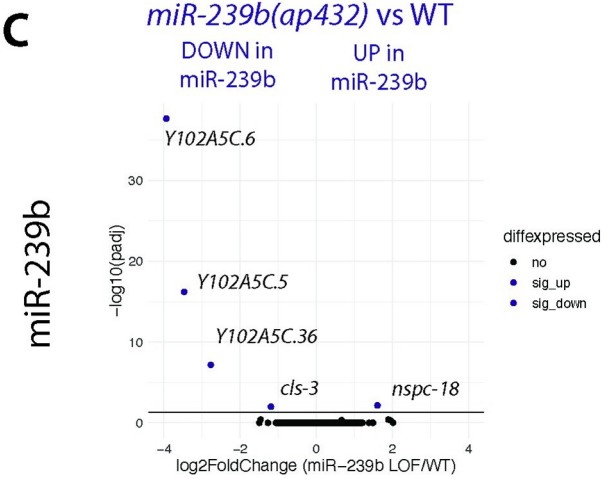

**Fig 3. Non-overlapping sets of genes are mis-regulated upon loss of each miR-238/239ab family member.** Volcano plots representing gene expression changes upon the loss of *miR-238(n4112)* (A), *miR-239a(ap439)* (B), and *miR-239b (ap432)* (C) compared to WT in day 5 adult *C. elegans* from three independent replicates. Colored dots (aqua for *miR-238*, gold for *miR-239a*, and purple for *miR-239b*) represent genes with a padj < 0.05 and baseMean >100. Top genes up- and down-regulated in each background are labeled.

**Table 1. Genes significantly up- and down-regulated in miR-238(-).** List of genes significantly up- and down-regulated in *miR-238(n4112)* compared to wildtype day 5 animals. Genes with previously reported longevity phenotypes and the supporting references are indicated [34–38].

| Common Name | Sequence Name | log2 (FC) | Description | RNAi Longevity Phenotype | Longevity References |
|---|---|---|---|---|---|
| ilys-3 | C45G7.3 | 2.22 | Invertebrate LYSozyme | n/a | |
| oac-54 | W07A12.6 | 1.97 | O-ACyltransferase homolog | n/a | |
| pqn-36 | F39D8.1 | 1.89 | Prion-like-(Q/N-rich)-domain-bearing protein | n/a | |
| clec-82 | Y54G2A.8 | 1.84 | C-type LECtin | n/a | |
| gsa-1 | R06A10.2 | 1.69 | G protein, Subunit Alpha | shortened life span | Xue H et al., 2007 |
| zip-8 | F23F12.9 | 1.64 | bZIP transcription factor family | n/a | |
| gtl-1 | C05C12.3 | 1.57 | Gon-Two Like (TRP subfamily) | n/a | |
| myo-5 | F58G4.1 | 1.50 | MYOsin heavy chain structural genes | n/a | |
| kin-36 | T22B11.4 | 1.38 | protein KINase | n/a | |
| | T10D4.6 | 1.35 | n/a | dauer lifespan extended in *daf-2(e1370); aak-1(tm1944); aak-2(ok524)* background | Xie M & Roy R, 2012 |
| tag-273 | Y57G11A.1 | 1.29 | n/a | n/a | |
| acs-2 | F28F8.2 | 1.27 | fatty Acid CoA Synthetase family | n/a | |
| cht-1 | C04F6.3 | 1.23 | CHiTinase | n/a | |
| | Y105C5B.3 | 1.22 | n/a | n/a | |
| | F56D5.6 | 1.22 | n/a | n/a | |
| lec-2 | F52H3.7 | 1.19 | gaLECtin | n/a | |
| oac-14 | F09B9.1 | 1.19 | O-ACyltransferase homolog | n/a | |
| clik-2 | C53C9.2 | 1.13 | CaLponIn-liKe proteins | n/a | |
| cest-12 | Y75B8A.3 | 1.12 | Carboxyl ESTerase domain containin | n/a | |
| nmy-1 | F52B10.1 | 1.12 | Non-muscle MYosin | n/a | |
| | C50F7.5 | 1.07 | n/a | n/a | |
| | T14G8.3 | 1.04 | n/a | n/a | |
| pqn-31 | F21C10.8 | 1.02 | Prion-like-(Q/N-rich)-domain-bearing protein | n/a | |
| acp-5 | F13D11.1 | 1.01 | ACid Phosphatase family | n/a | |
| hil-3 | F22F1.1 | 0.95 | HIstone H1 Like | n/a | |
| | ZK380.6 | -2.23 | | n/a | |
| | ZC317.7 | -2.13 | | n/a | |
| | C36C9.10 | -2.04 | | n/a | |
| grd-4 | T01B10.1 | -1.93 | groundhog (hedgehog-like family) | n/a | |
| | Y40A1A.6 | -1.82 | | n/a | |
| | C06A5.12 | -1.82 | | n/a | |
| irg-4 | F08G5.6 | -1.57 | Infection Response Gene | n/a | |
| | Y73F8A.1173 | -1.53 | | n/a | |
| | B0511.11 | -1.51 | | n/a | |
| | F57A10.4 | -1.49 | | n/a | |
| clec-52 | B0218.8 | -1.46 | C-type LECtin | n/a | |
| acbp-6 | Y17G7B.1 | -1.46 | Acyl-Coenzyme A Binding Protein | dauer lifespan extended | Xie M & Roy R, 2012 |
| spp-12 | T22G5.7 | -1.45 | SaPosin-like Protein family | shortened lifespan in *fer-15(b26);daf-2(mu150);fem-1 (hc17)* background | Murphy CT et al., 2003 |
| | F23D12.11 | -1.45 | | n/a | |
| impt-1 | Y52B11A.2 | -1.42 | MPacT rwd (RWD) domain containing homolog | extended lifespan | Ferraz RC et al., 2016 |
| tag-51 | K02F3.1 | -1.40 | | n/a | |
| | C10B5.1 | -1.39 | | n/a | |

*(Continued)*

**Table 1.** (Continued)

| Common Name | Sequence Name | log2 (FC) | Description | RNAi Longevity Phenotype | Longevity References |
|---|---|---|---|---|---|
| clec-47 | T09F5.9 | -1.39 | C-type LECtin | n/a | |
| ugtp-1 | ZK370.7 | -1.35 | UDP-Galactose Transporter Protein family | n/a | |
| | C06H5.14 | -1.35 | | n/a | |
| clec-3 | C41H7.7 | -1.33 | C-type LECtin | n/a | |
| | EEED8.13 | -1.28 | | n/a | |
| | T16H12.2 | -1.27 | | n/a | |
| | C16A3.2 | -1.27 | | n/a | |
| | T28A11.2 | -1.27 | | n/a | |
| | F54D5.4 | -1.26 | | n/a | |
| | C23H5.8 | -1.24 | | n/a | |
| | D1086.3 | -1.22 | | n/a | |
| crn-5 | C14A4.5 | -1.13 | Cell-death-Related Nuclease | extended lifespan in *eri-1(mg366)* background | Curran SP & Ruvkun G, 2007 |
| | K08D8.5 | -1.13 | | n/a | |
| phip-1 | F36A2.8 | -1.12 | Protein HIstidine Phosphatase | n/a | |
| | C17G10.13 | -1.11 | | n/a | |
| mff-2 | F55F8.6 | -1.07 | Mitochondrial Fission Factor | n/a | |
| | ZC412.3 | -1.05 | | n/a | |
| tpk-1 | ZK637.9 | -1.03 | Thiamine PyrophosphoKinase | n/a | |
| | ZK643.2 | -1.03 | | n/a | |
| cup-15 | F42A8.3 | -1.02 | Coelomocyte UPtake-defective | n/a | |
| bath-27 | F14D2.1 | -1.02 | BTB and MATH domain containing | n/a | |
| adbp-1 | VW02B12L.4 | -1.00 | ADR-2 Binding Protein | n/a | |
| lmd-1 | F43G9.2 | -1.00 | LysM Domain (peptidoglycan binding) protein | n/a | |
| | R04F11.5 | -0.98 | | n/a | |
| | B0304.4 | -0.97 | | n/a | |
| | EEED8.14 | -0.95 | | n/a | |
| | C35E7.8 | -0.92 | | n/a | |

rectal glands (Fig 4C). In contrast, expression from the _pmiR-239b::GFP reporter was more concentrated in the neurons and vulval cells (Fig 4D).

We also explored the relationship of the miR-238/239ab miRNAs to the Argonaute Like Gene proteins, ALG-1 and ALG-2. It was previously reported that ALG-1 and ALG-2 have differing spatial expression patterns in aging and play opposing roles in *C. elegans* longevity [41]. Thus, examining how miR-238, miR-239a, and miR-239b interact with ALG-1 and ALG-2 could inform on aging roles for these three miRNAs. We ranked miRNA association with ALG-1 and ALG-2 from day 5 RNA immunoprecipitation data [41], and performed small RNA-seq in day 5 *alg-1(gk214)* and *alg-2(ok304)* mutant strains (Fig 4A and S2 Table). All three miRNAs immunoprecipitate with ALG-1 and ALG-2 at levels relatively commensurate with their levels of detection in total smRNA-seq at day 5 of adulthood (Fig 4A). Despite this proportionate association with AGOs, the miR-238, miR-239a, miR-239b miRNAs have different sensitivities to the loss of *alg-1*: compared to WT, miR-238 is ~3-fold down in *alg-1 (gk214)*, while miR-239b is ~2-fold up, and there is no significant change for miR-239a (Fig 4A). No significant changes in these miRNAs were detected in *alg-2(ok304)* compared to WT

**A**

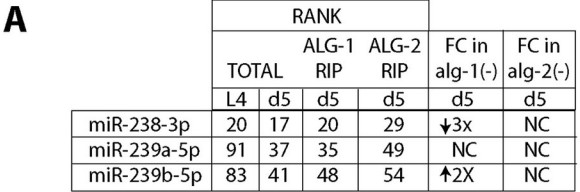

|  | RANK | | | | FC in alg-1(-) | FC in alg-2(-) |
|  | TOTAL | | ALG-1 RIP | ALG-2 RIP | | |
|  | L4 | d5 | d5 | d5 | d5 | d5 |
|---|---|---|---|---|---|---|
| miR-238-3p | 20 | 17 | 20 | 29 | ↓3x | NC |
| miR-239a-5p | 91 | 37 | 35 | 49 | NC | NC |
| miR-239b-5p | 83 | 41 | 48 | 54 | ↑2X | NC |

**B**

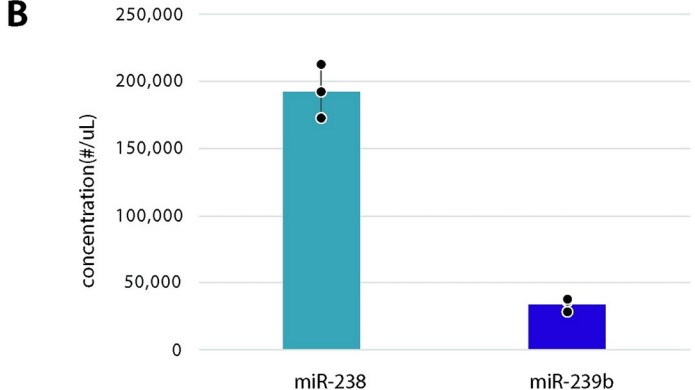

**C**

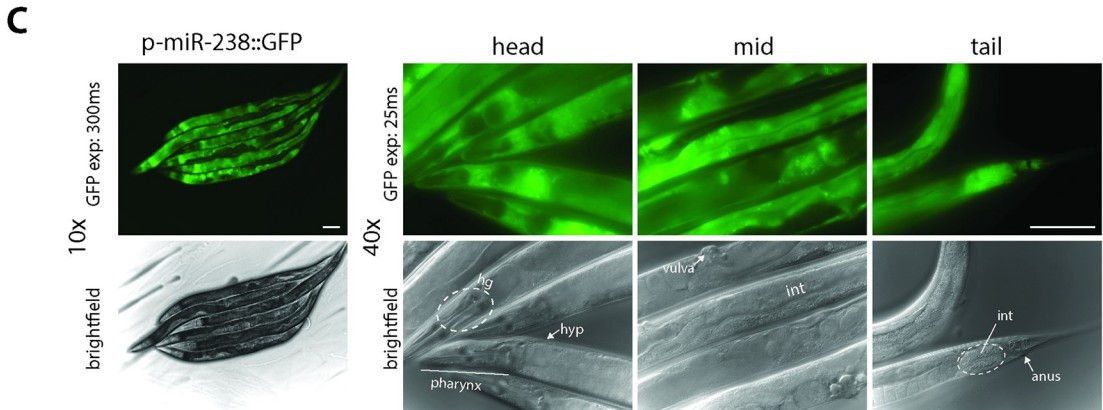

**D**

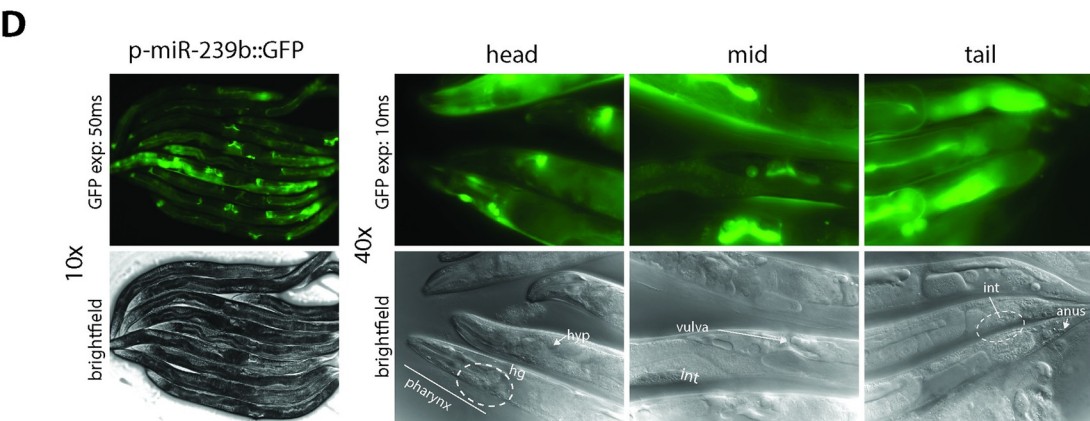

**Fig 4. Expression differences of miR-238, miR-239a, miR-239b microRNA family members.** (A) The levels of miR-238, miR-239a, and miR-239b detected in total RNA from L4 [39] and day 5 adults and in ALG-1 and ALG-2 RNA immunoprecipitates (RIP) from day 5 adults [41] are indicated by their rank compared to all other miRNAs detected with 1 being the most abundantly detected miRNA. Fold change (FC) in miRNA levels detected in day 5 *alg-1(gk214)* or *alg-2(ok304)* mutants compared to WT animals (n = 5 independent replicates of each strain, padj < 0.05 for significant FC). NC denotes no significant fold change. (B)

Absolute quantification of miR-238 (aqua) and miR-239b (purple) in day 5 WT samples. Bar graph represents mean of three biological replicates, individual replicate data indicated with dots. The error bars represent SDs. (C-D) Detection of microRNA expression patterns using GFP reporters fused to miR-238 (C) and miR-239b (D) promoter sequences. GFP fluorescence (*top*) taken at 10x of the whole body (300, 50 microsecond exposure) and 40x of the head, mid-section, and tail (25, 10 microsecond exposure) with accompanying DIC images (bottom). Select anatomical features are labelled, the vulva, intestine (int), anus, hypodermis (hyp), head ganglia (hg), and pharynx, white scale bar is 10uM.

day 5 animals (Fig 4A). In addition to the differences in quantity and spatial expression of these miRNAs, the variable sensitivity to loss of *alg-1* further shows that the expression and/or stability of miR-238, miR-239a, and miR-239b are subject to differential regulation.

## The longevity role of miR-238 can be replaced by miR-239a or miR-239b

Despite belonging to the same miRNA family, the loss of miR-238 causes a reduced lifespan with many transcripts mis-regulated, while the loss of miR-239a or miR-239b results in no apparent effect on lifespan and mis-regulation of very few genes (Figs 2A and 2B, and 3A-3C). These distinctions could be due to differences in miRNA expression (Fig 4) or in target RNA interactions due to nonidentical 3' end sequences (Fig 1C), or a combination of both. To investigate these possibilities, we used CRISPR/Cas9 to replace the endogenous miR-238 precursor sequence with that of miR-239a or miR-239b (Fig 5A). In these newly created strains, we detected increased levels of miR-239a and miR-239b specifically in the corresponding strains with the pre-miRNA replacing miR-238, consistent with expression from the miR-238 locus in addition to the endogenous gene (Fig 5B). The elevated expression of miR-239a or miR-239b from the *miR-238* locus resulted in no apparent effect on brood size or thermotolerance as compared to WT animals (S2A and S2B Fig).

We then asked if replacement of miR-238 with miR-239a or miR-239b would prevent the reduced lifespan caused by loss of miR-238. In lifespan analyses, the *pmiR-238::miR-239a* and *pmiR-238::239b* strains had survival curves indistinguishable from that of WT animals and were significantly longer lived than the *miR-238(n4112)* strain (Fig 5C). Overall, these data show that expression of miR-239a or miR-239b from the *miR-238* locus can rescue the reduced lifespan associated with loss of miR-238. Thus, differences in expression, and not the 3' end sequences, underlie the distinct role in aging of miR-238 compared to its sister miRNAs, miR-239ab.

## Discussion

Here, we investigated the individual roles of the miR-238/239ab miRNAs in adult *C. elegans*. We confirmed that the loss of miR-238 leads to a reduced lifespan but could not detect a longevity phenotype in animals lacking mature miR-239a or miR-239b. Additionally, the loss of individual or combined miR-238/239ab family members did not obviously impact fertility or thermoresistance. Consistent with the phenotypic observations, dozens of genes were mis-regulated *miR-238(-)* adults, while almost no changes in gene expression were detected in the *miR-239a* or *miR-239b* mutant strains. We found that the functional differences between miR-238 and its miR-239ab sisters result predominantly from expression and not sequence distinctions. Thus, the miR-238/239ab family of miRNAs positively regulates longevity through a mechanism that largely depends upon expression but not sequences beyond the seed region.

## The role of miR-238/239ab in stress and aging

Members of the miR-238/239ab family of miRNAs were among the first miRNAs proposed to regulate longevity in any organism [44]. Moreover, these miRNAs seemed to play opposing roles, as loss of miR-238 resulted in a shortened lifespan, while deletion of miR-239ab resulted

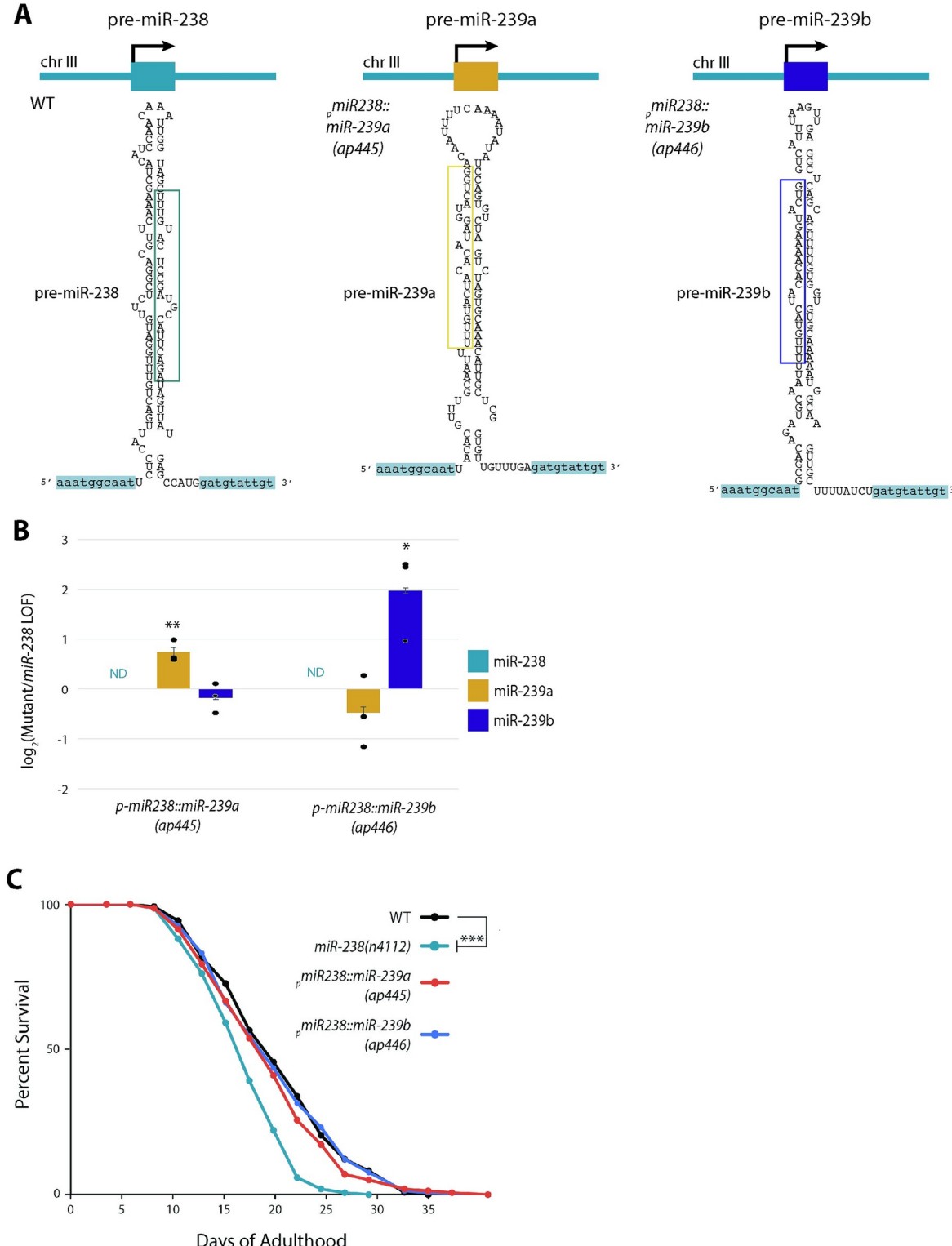

**Fig 5. The longevity role of miR-238 can be replaced by miR-239a or miR-239b.** (A) Schematic of the *miR-238* locus in WT (*pmiR-238::miR-238*) (top), in the *pmiR-238::miR-239a* strain (middle), and the *pmiR-238::miR-239b* strain (bottom). (B) TaqMan RT-qPCRs of miR-238, miR-239a, miR-239b mature miRNA levels in the *p*miR-238::miR-239a, and *p*miR-238::miR-239b replacement strains compared to the *miR-238(n4112)* background at day 5 of adulthood. The mean from 3 independent replicates is plotted; individual replicate data indicated with dots and error bars represent SDs. (C) Representative survival curves for WT (black), *miR-238(n4112)* (aqua), *pmiR-238::*

*miR-239a* (blue), and *pmiR-238::miR-239b* (coral) showing that the reduced lifespan due to loss of miR-238 is rescued by expression of miR-239a or miR-239b from the *miR-238* locus. *** P<0.0001 (log-rank).

in an extended lifespan and increased tolerance to heat and oxidative stress [28]. The best available reagent at the time of those studies was the *nDf62* strain, which has a 2,333 base pair deletion that removes both *miR-239a* and *miR-239b*, as well as an annotated non-coding RNA (*C34E11.9)* and small nucleolar RNA (snoRNA) (*C34E11.20)* (Fig 1A). Subsequent work confirmed enhanced thermal resistance but failed to reproduce a lifespan phenotype for the *nDf62* deletion strain [30,45]. It is currently unclear if the basis for the discrepancy in an aging phenotype for the *nDf62* strain is due to assay differences or unrecognized strain polymorphisms.

With the newer availability of gene editing tools, we were able to examine more precisely the roles of miR-239ab in aging and heat stress. In strains lacking expression of mature miR-239a, miR-239b or both miRNAs, no differences in lifespan or thermoresistance were detected when compared to WT animals. Likewise, almost no changes in gene expression were apparent in WT versus *miR-239a(-)* or *miR-239b(-)* day five adults. Thus, our work indicates that loss of the miR-239ab miRNAs is unlikely to contribute to any effects on longevity or thermal stress previously assigned to the *nDf62* strain.

Regardless of a functional role in aging, development or stress, miR-239ab have been consistently identified as miRNAs up-regulated under those conditions [28,30,39,46]. Additionally, a strain expressing GFP driven by the *miR-239* promoter was found to be a predictor of longevity; whether looking at GFP expression at day 3 or 7 post hatching, higher levels of GFP correlated with shorter lifespans in individual transgenic animals [43]. In a previous study, we identified a Heat Shock Element (HSE) bound by Heat Shock Factor 1 (HSF-1) that is situated between the *mir-239a* and *miR-239b* loci [39]. Additionally, we showed that increased expression of miR-239b upon heat shock is dependent on HSF-1 [39]. The increased expression of miR-239ab in aging animals could reflect the activity or availability of HSF-1. The Heat Shock Response (HSR), including the ability to up-regulate several Heat Shock Proteins (HSPs), declines abruptly in young *C. elegans* adults [47,48]. This event is due to formation of repressive chromatin at *hsp* promoters rather than obvious changes in HSF-1 levels or DNA binding ability [48]. It is possible that exclusion of HSF-1 from *hsp* genes in early adulthood allows for greater occupancy at other targets, including the HSE proximal to *miR-239ab*. Thus, the correlation between higher expression driven by the *miR-239* promoter and reduced life expectancy in individual *C. elegans* may reflect aberrant reprogramming of HSF-1 and/ or triggering of a stress response in young adults.

As previously reported, we observed that loss of miR-238 results in a markedly reduced lifespan [28]. A strain lacking expression of all three miR-238/239ab miRNA sisters phenocopies the shortened lifespan of *miR-238(-)* mutants, indicating that miR-238 alone regulates longevity. Loss of miR-238 or the entire miR-238/239ab family of miRNAs has no obvious impact on development or fertility in animals grown under standard lab conditions at 20˚C. Thus, the reduced lifespan of *miR-238(-)* mutants does not seem to be the result of general unhealthiness. The set of genes down-regulated in *miR-238(-)* at adult day 5 were enriched for defense response and immune system processes (S1 Table). This included *irg-4* which, when depleted by RNAi, leads to increased pathogen susceptibility, and *spp-12*, which encodes an antimicrobial polypeptide that promotes the extended lifespan of *daf-2* mutants with reduced insulin signaling [36,49–52]. Given that increased susceptibility to infection is linked to shortened lifespan and the innate immune system contributes to longevity [53–55], it is possible that down-regulation of innate immunity and pathogen susceptibility genes contributes to the *miR-238(-)* reduced lifespan. While dozens of genes were found to be mis-regulated in *miR-*

*238(n4112)* day 5 adults, none of them are predicted targets of this miRNA [32,33]. So far, the only validated target of miR-238 is the nicotinic acetylcholine receptor (nAChR), *acr-19* [56]. While mis-regulation of *acr-19* in *miR-238* mutant animals caused an abnormal nicotine withdrawal response [56], this role is unlikely to be related to the lifespan function of miR-238, as expression of this target was unchanged in *miR-238(n4112)* compared to WT day 5 adults.

## Redundant and distinct functions of miRNA family members

Considering the key role of the seed sequence in miRNA-target interactions [2], it is reasonable to expect that members of a miRNA family will have overlapping targets and, hence, functions. While this appears to be the case for some miRNA families, such as mir-35-42 and miR-51-56 in *C. elegans* [10–12], there are also examples of single family members having specific targets and roles [2]. Differences in 3' end sequences can bias target interactions to favor pairing with individual family members [14]. Cross-linking and immunoprecipitation with sequencing (CLIP-seq) assays that isolated chimeric sequences consisting of a target site ligated to a miRNA have revealed numerous instances of target occupancy restricted to a specific miRNA sister [15,18]. Favored binding affinities mediated by distinct 3' end sequences likely explain some of these specific miRNA-target interactions [15,18]. As the aging function of miR-238 could be replaced by miR-239a or miR-239b, sequence divergence among these miRNA sisters appears irrelevant for the distinct longevity role of miR-238 in adult *C. elegans*.

Differences in expression levels or domains can also lead to specific roles for miRNA family members [2]. Reporter-based and biochemical methods have shown that many *C. elegans* miRNAs, including some that belong to families, exhibit distinct temporal and spatial expression patterns [42,57–60]. Even miRNAs that are co-expressed as part of a mirtron can accumulate disproportionately, due to differences in processing and/or stability of the mature forms [12]. Here and in prior studies using reporter strains containing GFP fused to miRNA promoter sequences, expression of miR-238 appeared ubiquitous, whereas expression of miR-239 was primarily observed in neuronal and vulval cells in adult animals [28,42,43]. However, isolation of mature miRNAs directly, or as part of Argonaute complexes, from neuronal, pharyngeal, intestinal or body wall muscle cells did not reveal obvious differences in tissue specific accumulation of miR-238/239ab family miRNAs [57–60]. Despite this, the levels of miR-238, miR-239a, and miR-239b responded differently to the loss of *alg-1*: miR-238 was down-regulated, miR-239a unaffected, and miR-239b up-regulated. This suggests that the miR-238/239a/b family is differentially regulated by *alg-1*, either directly or indirectly. Even with these regulatory as well as spatial and total expression differences, we found that miR-239a or miR-239b could replace miR-238 function when expressed from the *miR-238* locus. These results suggest that regulatory elements in the *miR-238* gene control the expression of this sister miRNA in a manner needed for its longevity role.

In conclusion, this study corroborates that *miR-238* promotes longevity and contradicts a previously ascribed role for *miR-239ab* in limiting lifespan [28]. Moreover, we demonstrate that differences in expression, but not sequence, explain the inability of miR-239ab miRNAs to compensate for the loss of miR-238. This feature of the miR-238/238ab family may provide flexibility to maintain target regulation under conditions encountered in the wild that change the expression of individual sisters.

## Materials and methods

### Strain generation

CRISPR/Cas9 genome editing methods were used to generate the miR-239a and miR-239b LOF and the miR-238 replacement strains. PQ636 *miR-239a(ap439)*, PQ592 *miR-239b*

(*ap432*), PQ593 *miR-239b*(*ap433*) and PQ600 *miR-239a*(*ap435*), *miR-239b*(*ap432*) were generated by following methods described in Paix *et al*. with modifications suggested by the Dernburg lab [61]. Young adult wildtype worms (N2) were injected with mixes that included 0.5 uL of *dpy-10* crRNA (100uM), 1.0 uL of the appropriate crRNA, 2.5 uL of tracrRNA (100uM), and 7uL of Cas9 (40uM); Cas9 protein, tracrRNA, and crRNA were ordered from IDT. Worms were grown at 25˚C. 3 days later, *dpy* + *C. elegans* were singled onto new plates and PCR screened for disruptions in the *miR-239a* or *miR-239b* genes. To make the PQ679—*miR-238(ap445[PmiR-238::pre-miR-239a::miR-238 UTR] III);* and PQ680—*miR-238(ap446[PmiR-238::pre-miR-239b::miR-238 UTR] III);* strains, young adult wildtype worms (N2) were injected following methods for dsDNA asymmetric-hybrid donors as described in Dokshin *et al*. [62]. The injection mix included 5μg Cas9 protein, 2mg tracrRNA, 1.12μg crRNA, 800ng pRF4::rol-6 plasmid and 4μg of a dsDNA donor cocktail. Homology arms were 120bp long. F1 rollers were singled onto new plates as well as non-roller siblings from the same plate. After laying progeny, F1 were lysed and PCR screened for integration of the pre-miR-239a or pre-miR-239b sequence into the *miR-238* locus. Editing was confirmed by Sanger sequencing and successful mutant strains were backcrossed at least three times to N2. All strains and oligonucleotide sequences are listed in S3 Table.

In Vivo Biosystems was contracted to generate the ₚ*miR-239b*::*GFP* strain (COP2506). Using MosSCI transgenesis methods [63], genetic cargo containing *pmir-239.b*::*GFP* and an *unc-119* rescue cassette, was injected for insertion into the *ttTi5605 Mos1* locus on chr2 in the *C. elegans* genome. Candidate lines were screened for rescue of function on the *unc-119(ed3) III* mutant allele and insertion confirmed by genotyping.

## Nematode culture and lifespan analyses

*C. elegans* strains were cultured under standard conditions and synchronized by hypochlorite treatment. Lifespan analyses were conducted at 20˚C in the absence of FUdR, as previously described [64]. Embryos were plated on NGM plates containing OP50 and the first day after the L4 stage was regarded as adult day 0. Worms were picked onto fresh food every other day until reproduction ceased and scored for viability every 2 to 3 days. Animals that died by bagging, bursting, or crawling off the plates were censored. JMP IN 16 software was used for statistical analysis and P-values were calculated using the log-rank (Mantel-Cox) method. Lifespan assays were performed in a blinded manner and statistics for all replicates (3–12 independent) are shown in S4 Table.

## Brood size assays

Between 5–9 individual L4 *C. elegans* of each genotype were moved to individual plates seeded the day prior with OP50. Every day post adulthood, the parental adult *C. elegans* was moved to a new plate, and the eggs were counted. This was done until the end of the reproductive span of the individual animal. Brood size assays were performed in a blinded manner and data for all replicates (3 independent) are shown in S4 Table.

## Thermotolerance assays

Adult heat shock experiments were carried as described in De Lencastre *et al*. and Nehammer *et al*. with minor alterations, such as not using FuDr to stop progeny production [28,30]. For the de Lencastre *et al*. thermotolerance protocol: *C. elegans* strains were cultured under standard conditions and synchronized by hypochlorite treatment [65]. Heat shock viability assays were performed by plating bleach synchronized L1 worms rocked at 20˚C overnight on UV treated small NGM plates seeded with OP50 the day before. Worms were grown until L4, then

for an additional 36 hours at 20˚C before raising the temperature to 35˚C for 12 hrs of heat shock. Assays were blinded before heat shock and were unblinded only after scoring viability. For the Nehammer *et al*. thermoresistance protocol: Gravid adults were allowed to egg lay for a 2-hour period to produce relatively synchronized populations of progeny at 20˚C on UV treated NGM plates seeded with OP50 the day before. From the mid-point of the egg lay, worms were grown for 86 hours, and during the first day of adulthood worms were moved to new UV treated small NGM plates seeded with OP50 the day before. Those adult worms were then incubated at 35˚C for 12 hrs of heat shock. Worms recovered for 24hrs at 20˚C before scoring. For all heat shock experiments, at least 100 worms were subjected to heat shock or control 20˚C conditions per strain per replicate with no more than 20 worms per single small NGM plate. Thermoresistance assays were performed in a blinded manner and data for all replicates (3 independent) are shown in S4 Table.

## qRT-PCR

RT-PCR analyses of miRNA (TaqMan) levels were performed according to manufacturer's instructions with the StepOnePlus and QuantStudio 3 Real-Time PCR Systems (Applied Biosystems). Levels were normalized to U18 snoRNA. Three biological replicates were performed with three technical replicates for each target gene. Numerical data are provided in S5 Table. Not detected (ND) was called for samples that were flagged as inconclusive or no amplification in all three biological and three technical replicates in the QuantStudio 3 Software (Thermofisher).

Absolute quantification of miRNA levels was performed by generating a standard curve to estimate the absolute copy number of the target miRNA in biological samples [66,67]. Standard curves were generated using 10-fold dilutions, $10^7$–$10^1$ copy/uL, of synthetic RNA oligonucleotides ordered from IDT that were identical in sequence to miR-238-3p, and miR-239b-5p. In parallel, three biological replicates of adult day 5 wildtype (N2) were assayed for miR-238-3p, and miR-239b-5p expression (TaqMan). The standard curve fit equation was used to approximate copies of endogenous miRNA from the Ct values obtained in the biological samples.

## RNA sequencing

RNA was isolated from wildtype (N2), *miR-238(n4112)*, *miR-239a(ap439)*, and *miR-239b (ap432)* day 5 adults. Adult *C. elegans* were separated from eggs and progeny daily by washing plates with M9 into conical tubes and allowing the adults to settle by gravity for a few minutes on a bench top. The supernatant containing larvae and eggs was then removed, and this process was repeated 3–15 times until eggs and larvae were no longer visible. Three independent poly(A) selected RNA-seq libraries were prepared from each strain for sequencing with the Illumina TruSeq mRNA Library Prep Kit. cDNA libraries were sequenced on an Illumina NovaSeq 6000. Libraries were at least 24 million reads per sample. Reads were aligned to the *C. elegans* genome WBcel282 assembly using STAR and the average percent of uniquely mapped reads was 94% [68]. Aligned reads were sorted using Samtools [69] and reads were then quantified using featureCounts using WBcel282 gene annotations [70]. Differential expression was calculated using DESeq2 [71]. Genes with a basemean of at least 100, and adjusted p-value of $< 0.05$ were considered significantly mis-regulated in mutant versus wild type animals (S1 Table). Volcano plots were generated using ggplot2 in R [72,73]. Enrichment analysis was performed using WormBase's enrichment analysis tool [74,75].

## smRNA-seq

Small RNA sequencing was performed on five independent replicates of synchronized wild-type (N2), *alg-1(gk214)*, and a*lg-2(ok304)* strains collected on day 5 of adulthood. Strains were

cultured at 20°C to day five of adulthood and collected for RNA isolation. Eggs and progeny were separated from adult worms through daily washes with M9 solution, followed by gravity separation of pelleted adult worms from the supernatant containing eggs and progeny. The supernatant was aspirated and M9 washed, this was repeated until the M9 remained clear. Total RNA was isolated and smRNA libraries were then prepared from 1 μg of total RNA using the Illumina TruSeq Small RNA Library Prep Kit. Once prepared, smRNA libraries were sent for single-end sequencing on an Illumina HiSeq 4000. Adapter sequences were removed using Cutadapt, and smRNA reads were mapped to the annotated *C. elegans* genome (WS266) using Bowtie-build to first create indices and miRDeep2 to align and quantify reads [76,77]. Differential expression analysis was performed by first normalizing reads to library size (read counts per million) and then measuring the log2foldchange of mutants to WT strains within replicates. MiRNAs were called significantly misregulated if they exhibited an absolute mean log2foldchange greater than 1.5 and a padj less than 0.05 in mutant versus wildtype samples. The results are summarized in S2 Table.

## Supporting information

**S1 Table. related to Fig 3.** Differentially expressed genes in *miR-238(n4112)*, *miR-239a (ap439)*, or *miR-239b(ap432)* mutants compared to WT at day 5 of adulthood and their tissue, phenotype, and gene enrichment terms. (Spreadsheet uploaded separately). (XLSX)

**S2 Table. related to Fig 4.** Differentially expressed miRNAs in *alg-1(gk214)* or *alg-2(ok3034)* mutants compared to WT at day 5 of adulthood. (Spreadsheet uploaded separately). (XLSX)

**S3 Table. Lists of strains and oligonucleotide sequences used in this study (Spreadsheet uploaded separately).** (XLSX)

**S4 Table. related to Figs 2A–2F and 5E, and S2A–S2B.** Statistics of all lifespan, brood size and thermoresistance assays used in this study. (Spreadsheet uploaded separately). (XLSX)

**S5 Table. related to Figs 1D, 4B and 5B.** Numerical data underlying graphs and summary statistics for qRT-PCR. (Spreadsheet uploaded separately). (XLSX)

**S1 Fig. Predicted structures of *miR-239a* and *miR-239b* precursors in wildtype and mutant strains.** RNAfold structures for wildtype pre-miRNA structures of miR-239a and miR-239b have the mature miRNA sequence boxed in yellow and blue, respectively. New loss of function mutants generated in this study by CRISPR/Cas9 are indicated with mature miRNA sequences boxed in red. *miR-239a(ap439)* deletes 10 nucleotides in the 3' arm of the stem. *miR-239a (ap435)* inserts 25 nucleotides into this region. *miR-239b(ap432)* deletes 15 nt at the base of the 3' arm of the stem and *miR-239b(ap433)* deletes the GCAAAAA sequence and inserts 26 nt. (PDF)

**S2 Fig. Brood size and heat shock survival for $_p$miR-238::miR-239a and $_p$miR-238::miR-239b.** (A) Results from brood size analysis; *miR-238(n4112)* (aqua), $_p$miR-238::miR-239a (coral), and $_p$miR-238::miR-239b (blue) do not have a statistically significant difference when compared to WT (black). ANOVA and the post hoc test (Tukey's HSD). Bar graph represents mean of three biological replicates; individual replicate data indicated with black dots. The

error bars represent SDs. (B) Results from heat shock on day 2 adults for 15 hours at 32˚C followed by recovery for 24hr at 20˚C. *miR-238 (n4112)* (aqua), *pmiR-238::miR-239a* (coral), and *pmiR-238::miR-239b* (blue) do not have a statistically significant different percent heat shock survival when compared to WT (black). ANOVA and the post hoc test (Tukey's HSD). Bar graph represents mean of three biological replicates; individual replicate data indicated with black dots. The error bars represent SDs.
(PDF)

## Acknowledgments

We thank members of the Pasquinelli Lab for helpful discussions and critical reading of the manuscript.

## Author Contributions

**Conceptualization:** Laura B. Chipman, Amy E. Pasquinelli.

**Data curation:** Laura B. Chipman, San Luc, Ian A. Nicastro, Jesse J. Hulahan, Delaney C. Dann, Devavrat M. Bodas, Amy E. Pasquinelli.

**Formal analysis:** Laura B. Chipman, San Luc, Ian A. Nicastro, Amy E. Pasquinelli.

**Funding acquisition:** Amy E. Pasquinelli.

**Investigation:** Laura B. Chipman, San Luc, Ian A. Nicastro, Jesse J. Hulahan.

**Methodology:** Laura B. Chipman, San Luc, Ian A. Nicastro, Jesse J. Hulahan, Delaney C. Dann, Devavrat M. Bodas.

**Project administration:** Amy E. Pasquinelli.

**Supervision:** Amy E. Pasquinelli.

**Validation:** Laura B. Chipman, San Luc, Ian A. Nicastro, Jesse J. Hulahan, Delaney C. Dann, Devavrat M. Bodas, Amy E. Pasquinelli.

**Visualization:** Laura B. Chipman, San Luc, Ian A. Nicastro, Amy E. Pasquinelli.

**Writing – original draft:** Laura B. Chipman.

**Writing – review & editing:** Laura B. Chipman, San Luc, Ian A. Nicastro, Jesse J. Hulahan, Delaney C. Dann, Amy E. Pasquinelli.

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
