## [Decision Letter · Decision Letter 0]

28 Jul 2023

Dear Dr Pasquinelli,

Thank you very much for submitting your Research Article entitled 'Expression, not sequence, distinguishes miR-238 from its miR-239ab sister miRNAs in promoting longevity in Caenorhabditis elegans' to PLOS Genetics.

The manuscript was fully evaluated at the editorial level and by independent peer reviewers. The reviewers appreciated the attention to an important problem, but raised some substantial concerns about the current manuscript. Based on the reviews, we will not be able to accept this version of the manuscript, but we would be willing to review a much-revised version. We cannot, of course, promise publication at that time.

If you decide to revise the manuscript for further consideration at PLOS Genetics, please aim to resubmit within the next 60 days, unless it will take extra time to address the concerns of the reviewers, in which case we would appreciate an expected resubmission date by email to plosgenetics@plos.org.

We are sorry that we cannot be more positive about your manuscript at this stage. Please do not hesitate to contact us if you have any concerns or questions.

Yours sincerely,

Eric A Miska, PhD

Academic Editor

PLOS Genetics

Gregory Copenhaver

Editor-in-Chief

PLOS Genetics

Reviewer's Responses to Questions

**Comments to the Authors:**

Reviewer #1: The manuscript by Pasquinelli and colleagues addresses important questions in the fields of miRNA-based gene regulation and miRNA-dependent lifespan control by using precise CRISPR/Cas9 editing of miRNA loci in C. elegans.

miRNA-dependent lifespan control. Using more advanced genome editing techniques the authors re-access previous conclusions claiming the opposite roles of the same miRNA family members miR-238 and miR-239a/b in lifespan regulation. They show that miR-238 contributes to the normal lifespan of C. elegans, consistent with previous results. However, their conclusions regarding miR-239a and miR-239b do not support earlier studies suggesting the role of these miRNAs in reducing lifespan (i.e. longer lifespan of the mutants). The lifespans of several miR-239a or miR-239b loss-of-function mutant alleles as well as that of miR-239a; miR-239b double are not different from the wild type. These results are very convincing. Since the earlier work used a deficiency allele removing ncRNA genes, in addition to miR-239a and miR-239b, it would be interesting for the field to clarify the molecular mechanism responsible for the lifespan extension of the deficiency strain.

miRNA-based gene regulation. miR-238 shares the seed region with miR-239a and miR-239b but has a distinct 3’-end sequence. The authors investigate the importance of this difference in sequence for lifespan regulation by miR-238, and not miR-239a/b. In an elegant experiment, they replace the miR-238 sequence in its native locus with either miR-239a or miR-239b, demonstrating that these miRNAs can perform the miR-238 function. Therefore, a higher level of miR-238 expression and/or its tissue-specific expression is critical for its function.

Overall, the manuscript significantly contributes to the miRNA and longevity field and is suitable for publication in PLoS Genetics.

Suggestions for improving content presentation and clarity:

1. In Figure 3, the right panels listing the top misregulated genes in the mutants convey the point of more genes changing expression in the absence of miR-238 but otherwise are not informative. The authors briefly mention a lack of specific functional enrichments among the changed genes but do not show any Gene Set Enrichment Analyses. These can be provided, possibly as Supplemental information.

I also suggest including a Table listing 25 significantly upregulated and 44 sign. downregulated genes in miR-238(-) in the main text, with names and short descriptions based on WormBase, especially noting any connection to lifespan regulation. This will be useful for readers studying lifespan.

2. A more extended description of the connection to the previously published HSF-1 work from the lab and to the miR-239 promoter-driven GFP predictor of longevity will be useful for the reader. I find reading this part of the discussion difficult without pulling up the earlier papers. I assume that miR-239p::GFP expression in young adults or during development (when exactly?) predicts the animals’ lifespan.

Other issues:

Define NC in Figure 4A legend. Indicate specific anatomical structures where miR-238 is expressed higher than miR-239 between Figures 4B and 4C, e.g. vulva, etc.

Reviewer #2: In this manuscript, Pasquinelli and colleagues disentangle the roles of miR-238 and miR-239 in longevity. miR-238 shares a seed sequence with two closely related miRNAs, miR-239a and miR239b. In earlier work, loss of miR-239 was shown to extend lifespan while loss of miR-238 shortened lifespan, which was somewhat surprising given their sequence similarity. However, using a more thorough and precise approach involving genome editing, the authors show that disrupting either or both mir-239 genes has no affect on lifespan, while confirming that the mir-238 gene is indeed required for normal longevity. Furthermore, using miRNA hairpin swaps, the authors demonstrate the sufficiency of the miR-238/miR239 seed sequence for target recognition and show that differences in expression of these miRNAs, rather than differences in sequence, is responsible for their distinct roles . The results provide a thorough reassessment of the roles of three miRNA genes in aging, correct inaccuracy in earlier work, and further demonstrate the critical role the seed sequence plays in miRNA-target recognition. As such, the work is likely to appeal to a broad audience.

Specific Points

Lines 225-231. As I’m sure the authors are aware, RNA ligation is sensitive to sequence context. It doesn’t appear that degenerate bases were included in the small RNA library adapter sequences to minimize biases in ligation, and thus discussion of the total abundance of particular miRNAs should note this caveat since some miRNAs will be more efficiently captured than others.

Can the authors comment on possible roles for C34E11.9 or C34E11.20 in lifespan since these two genes are also deleted in the nDf62 strain and seemingly could also underlie the extended lifespan in this strain?

-Tai Montgomery

Reviewer #3: The manuscript by Chipman et al., “Expression, not sequence, distinguishes miR-238 from its miR-239ab sister miRNAs in promoting longevity in Caenorhabditis elegans” tackles an important outstanding question of functional differences among miRNAs belonging to the same family. The authors use miR-238, miR-239a, and miR-239b family miRNAs as a case in point for what likely is a repeated pattern of functional redundancy (or lack thereof) among miRNAs across animal development. Previous observations of opposite phenotypes among the individual members of this miRNA family raised questions about how this is achieved, considering the expected ability of these miRNAs to target same or overlapping target genes. Thus, this manuscript is a welcome analysis of functional roles of mir-238, mir-239a, and mir-239b, made possible by the authors’ generation of individual miRNA mutations in each family member locus. The authors then re-examine the previously published phenotypes to nail down the functional differences among the individual miRNAs. The most interesting finding of this paper is that miR-239a or miR-239b can functionally replace miR-238, suggesting that the distinct expression pattern between these miRNA family members likely drives the functional differences rather than sequence-based target specificity. This is a valuable finding that will inform how we interpret future functional miRNA redundancies or lack thereof. Given the fairly widespread unique spatial and temporal miRNA expressions, the reported observation is likely to become a commonly observed pattern and thus would set an important example of how unique functions among miRNA family members are achieved.

Overall, the manuscript is a great contribution to our understanding of non-redundancy among family miRNAs. However, the conclusion that “differences in expression” between mir-239a/b and mir-238 drive distinct functions is muddled by the current lack of clarity on relative expression levels and spatial expression patterns of the miRNAs in question. The manuscript would thus benefit from addressing the following:

The first concern centers on the relative abundances of the miRNAs. According to some published data, there is an apparent strong abundance disparity between mature miR-238 and miR-239a/b levels, with miR-239a/b being hundreds- if not a thousand-fold less abundant than miR-238. Figure 5B shows an ~2x fold increase in the amount of miR-239a present when miR-239a replaces miR-238, an increase seemingly concordant with its expression from a second locus. However, a 2-fold increase (or even the 4-fold increase, in case of miR-239b) over initial, very low miR-239 levels does not come close to reaching the levels of mature miR-238. How could a potentially small increase in miRNA levels replace loss of highly abundant miR-238? A more careful quantification of the mature miRNAs is needed, perhaps using curves generated with synthetic miRNAs to quantify “absolute” levels of miR-239a, b, and miR-238. Overall, more information on miRNA abundances is needed, along with an expanded discussion of how miR-239, a or b, can functionally replace miR-238, given the likely discrepancy in levels. Of course, additional probing of relevant miRNA levels could reveal that abundance difference is not as significant as previously reported, but this needs to be backed up by data. Currently, it is impossible to tell the overall levels of each of the miRNAs. If there is a significant discrepancy in miRNA abundances, then an extensive discussion should be included on how an inabundant miRNA can functionally replace its highly abundant “sister”.

It is also possible that it is the spatial expression that drives the differences. If the authors believe that the spatial expression contributes to lack of redundancy between miR-238 and miR-239a/b, additional careful analysis of the GFP expression is warranted. From the images, it looks like there may be some intestinal expression of miR-239b? Additional imaging of the tissues would be more convincing to the readers of the similarities and differences between the GFP expression patterns driven by the two loci promoters. (Please include scale bars on images as needed.)

These above questions have important bearing on the interpretation of the presented data and I believe could be, and should be addressed by the authors.

Additional comments:

Figure 1 and the associated text. Please depict/describe the generated mutations in greater detail. The text should clearly state which mutations are deletions (and what they delete) and which mutations are insertions or indels. As the reader reads the text, there is a bit of a discongruity between the text and the figure because the text does not give the reader a heads up on the fact that two of the mutations are insertions/indels. A short statement on why insertions or indels are deemed to be loss of function would be helpful. A predicted RNA fold of each resultant mutation would also be important to allow the readers assess the likelihood of each mutant producing a functional miRNA. For example, it appears that there are some differences between ap439 and ap435 (Figure 1D and Figure 2D), which could perhaps be due to the ability of ap435 produce a small amount of mature miRNA. A predicted RNA fold would thus be illuminating. Ideally, there should be a demonstration that ap439 and ap435 are equivalent loss of function alleles, perhaps through more accurate quantification methods (either smRNA-seq or using Taqman quantification in conjunction with synthetic miRNA-based curves). There are several other ways this could be addressed: ap435 could be generated as a single mutant, or generation of ap439 with ap432. At a minimum, differences between alleles should be clearly discussed within the context of the data and the potential impact on data interpretation (ex. if ap435 really is a weaker allele compared to ap439, some of the interpretations would change).

In addition, I would suggest drawing a diagram for each individual mutation, rather than combining two per diagram. It’s pretty effort-intensive for the reader to understand the structure of each individual allele from the figure legend. As the figure stands, the details on the exact nature of the alleles are missing (for example, ap432 deletion is not shown fully, neither is ap433 deletion). A more detailed schematic on each individual allele is needed.

Figure 1 shows mir-239b first, then mir-239a, while the legend describes mir-239a first. Please fix to align the order.

Figure 1D. See comment above. To repeat, because the double mutant uses ap435 in conjunction with ap432, the effects of ap435 alone on mir-239a expression should be shown of the allele difference/lack thereof should be addressed or at the very least more clearly discussed in text.

Figure 4A. To fully understand the presence or absence of functional redundancy of mir-238/239 miRNAs, it’s important to have a perspective on their relative abundances. I think using the ranking system as opposed to read counts obscures the important fact that miR-239a/b are seemingly quite inabundant compared to miR-238. The use of ranking is not justified by the authors, and it’s unclear what information Figure 4A provides in terms of understanding miR-238/239 relationship with AGOs. The conclusion that there is “differential regulation” is somewhat unsatisfactory. An expanded discussion of the data is warranted. Is the conclusion that mir-239a/b are not loaded into ALG-1 or ALG-2? Is the two-fold upregulation of mir-239b in alg-1(-) significant or biologically meaningful? Again, fold change differences without knowledge of levels are difficult to interpret.

Figure 5A is not very descriptive. Clarity could be improved by showing the actual sequence of the replacement, highlighting the replaced vs. endogenous locus sequence. Otherwise, the reader is left wondering exactly what was replaced. I think it’s stated once that this is a precursor miRNA replacement in Methods, but this point is easily missed and the readers would benefit from knowing the exact nature of the replacement.

Figure 5C-D: It is unclear what these panels contribute to the narrative of the paper. Since mir-238 mutant does not have a brood size or heat shock phenotype, the data shown do not provide any information on the ability of mir-239a/b replace mir-238 function. At best, the data demonstrate that the miRNA replacement does not produce an unexpected effect on brood size or heat stress response. As such, it should be relegated to supplemental data and the Results section’s language adjusted to more accurately reflect the information provided by these experiments.

Gene expression profiles in miRNA loci mutants were limited to day 5 old adults. Thus, care should be used to draw conclusion about miRNA functions and their targets (line 339 and on). Day 5 old adults is a single snapshot into gene expression and could have missed direct miR-238 targets, affected earlier in development/aging, but nonetheless having an effect of animal longevity.

smRNAseq: line 484-485. Is that 5 independent replicates for each of the strains or 5 replicates for all of the strains? It’s unclear as written.

**Have all data underlying the figures and results presented in the manuscript been provided?**

Reviewer #1: Yes

Reviewer #2: Yes

Reviewer #3: Yes

PLOS authors have the option to publish the peer review history of their article (what does this mean?). If published, this will include your full peer review and any attached files.

Reviewer #1: No

Reviewer #2: **Yes: **Taiowa Montgomery

Reviewer #3: No

---

## [Decision Letter · Decision Letter 1]

7 Nov 2023

Dear Dr Pasquinelli,

We are pleased to inform you that your manuscript entitled "Expression, not sequence, distinguishes miR-238 from its miR-239ab sister miRNAs in promoting longevity in Caenorhabditis elegans" has been editorially accepted for publication in PLOS Genetics. Congratulations!

Yours sincerely,

Eric A Miska, PhD

Academic Editor

PLOS Genetics

Gregory P. Copenhaver

Editor-in-Chief

PLOS Genetics

Comments from the reviewers (if applicable):

Reviewer's Responses to Questions

**Comments to the Authors:**

Reviewer #1: I am satisfied with the revised version of the manuscript.

Reviewer #2: The authors addressed my concerns and I believe the manuscript is ready for publication.

Reviewer #3: I still have a nagging concern regarding the nature of the various alleles. The fact that the mature miRNA sequences remain intact, and that remaining precursor miRNA still appears to fold into a decent hairpin (for some of the mutations, Supp Figure 1), leaves open the possibility that this can result in alternative duplex processing. This could produce a mature miRNA that is still able to function yet be sufficiently different at the 3’ end to evade accurate detection by Taqman, resulting in loss of detection shown in Figure 1D. Thus, I believe it is important to be completely transparent about the nature of the alleles with the reader.

The added statement that some of the alleles are deletions and some insertions (lines 147-151) provides no additional information on the exact nature of the alleles. As currently described, the wording is vague enough that it obscures the fact that the mature sequences of the miRNAs remain intact in the generated mutants. I believe it’s important to state that information clearly in conjunction with the added statement on lines 147-151, so that the reader can make their assessments of the data. In addition, it appears that Figure 1 has ample white space to accommodate 2 additional diagrams so that each individual allele gets a diagram, which would again allow the reader to better understand the nature of the alleles. These are small changes that can be easily implemented by the authors but would make a significant impact for the readers who may wish to remain cautious about the above-mentioned possibility and interpret the data accordingly.

For absolute quantifications, please show the standard curves in Supplemental data.

Figure 5A: there is no top, middle, and bottom—please correct legend to “left, middle, and right”, or separate into panels A-C.

**Have all data underlying the figures and results presented in the manuscript been provided?**

Reviewer #1: Yes

Reviewer #2: Yes

Reviewer #3: Yes

PLOS authors have the option to publish the peer review history of their article (what does this mean?). If published, this will include your full peer review and any attached files.

Reviewer #1: No

Reviewer #2: **Yes: **Taiowa Montgomery

Reviewer #3: No

**Data Deposition**

http://datadryad.org/submit?journalID=pgenetics&manu=PGENETICS-D-23-00563R1

**Press Queries**

---

## [Editor Report · Acceptance letter]

20 Nov 2023

PGENETICS-D-23-00563R1 

Expression, not sequence, distinguishes miR-238 from its miR-239ab sister miRNAs in promoting longevity in Caenorhabditis elegans 

Dear Dr Pasquinelli, 

We are pleased to inform you that your manuscript entitled "Expression, not sequence, distinguishes miR-238 from its miR-239ab sister miRNAs in promoting longevity in Caenorhabditis elegans" has been formally accepted for publication in PLOS Genetics! Your manuscript is now with our production department and you will be notified of the publication date in due course.

With kind regards,

Zsofi Zombor

PLOS Genetics

On behalf of:
